# Monitoring of Antimicrobial Drug Chloramphenicol Release from Electrospun Nano- and Microfiber Mats Using UV Imaging and Bacterial Bioreporters

**DOI:** 10.3390/pharmaceutics11090487

**Published:** 2019-09-19

**Authors:** Liis Preem, Frederik Bock, Mariliis Hinnu, Marta Putrinš, Kadi Sagor, Tanel Tenson, Andres Meos, Jesper Østergaard, Karin Kogermann

**Affiliations:** 1Institute of Pharmacy, Faculty of Medicine, University of Tartu, Nooruse 1, 50411 Tartu, Estonia; liis.preem@ut.ee (L.P.); andres.meos@ut.ee (A.M.); 2Department of Pharmacy, University of Copenhagen, Universitetsparken 2, DK-2100 Copenhagen Ø, Denmark; frederik.bock@sund.ku.dk; 3Institute of Technology, Faculty of Natural Sciences, University of Tartu, Nooruse 1, 50411 Tartu, Estonia; mariliis.hinnu@ut.ee (M.H.); marta.putrins@ut.ee (M.P.); kadisagor@gmail.com (K.S.); tanel.tenson@ut.ee (T.T.); 4LEO Foundation Center for Cutaneous Drug Delivery, Department of Pharmacy, University of Copenhagen, Universitetsparken 2, DK-2100 Copenhagen Ø, Denmark

**Keywords:** antibacterial activity, bacterial bioreporters, drug release, electrospinning, microfibers, nanofibers, UV imaging

## Abstract

New strategies are continuously sought for the treatment of skin and wound infections due to increased problems with non-healing wounds. Electrospun nanofiber mats with antibacterial agents as drug delivery systems provide opportunities for the eradication of bacterial infections as well as wound healing. Antibacterial activities of such mats are directly linked with their drug release behavior. Traditional pharmacopoeial drug release testing settings are not always suitable for analyzing the release behavior of fiber mats intended for the local drug delivery. We tested and compared different drug release model systems for the previously characterized electrospun chloramphenicol (CAM)-loaded nanofiber (polycaprolactone (PCL)) and microfiber (PCL in combination with polyethylene oxide) mats with different drug release profiles. Drug release into buffer solution and hydrogel was investigated and drug concentration was determined using either high-performance liquid chromatography, ultraviolet-visible spectrophotometry, or ultraviolet (UV) imaging. The CAM release and its antibacterial effects in disc diffusion assay were assessed by bacterial bioreporters. All tested model systems enabled to study the drug release from electrospun mats. It was found that the release into buffer solution showed larger differences in the drug release rate between differently designed mats compared to the hydrogel release tests. The UV imaging method provided an insight into the interactions with an agarose hydrogel mimicking wound tissue, thus giving us information about early drug release from the mat. Bacterial bioreporters showed clear correlations between the drug release into gel and antibacterial activity of the electrospun CAM-loaded mats.

## 1. Introduction

Electrospinning is a highly versatile and robust technique that allows production of fibers with diameters from several nanometers to tens of micrometers [1]. Drug-loaded electrospun nanofiber mats have been studied intensively and show potential as drug delivery systems (DDSs) [2] and tissue engineering scaffolds [3,4] due to several advantages, such as huge specific surface area, porosity and the possibility to modify the drug release kinetics [5,6,7]. Compared to other delivery systems, nano- and microfiber mats enable control and tuning of the drug release kinetics [8] and, hence, design the mats with desired properties [9,10]. For example, for local antibiotic delivery, the desired drug release needs to follow two steps: initial fast release followed by the slow zero-order kinetics over a longer period of time [11]. Novel strategies for attaining sustained release have been proposed, for example via the formation of core-shell structures [12,13], beads [14], or modification of nanofiber mat thickness [15]. The drug release process is affected by several factors, such as the physicochemical properties of the drug and carrier polymer, the structural characteristics of the material system, release environment, and the possible interactions between these factors [16]. It is known that drug release from electrospun fiber mats may vary depending on the material properties and the structure of the mats [9,15,17,18].

Despite the substantial body of literature on electrospun fiber mats and their characterization, there is no standard method for the analysis of drug release from fiber mats. Traditional pharmacopoeial drug dissolution tests have been found useful for the analysis of nanofibers incorporated into capsules or pressed into a tablet [19,20]. However, when electrospun nanofiber mats are intended for the local delivery of drug, e.g., wound therapy, the amount of available liquid is low. Thus, mimicking the actual biorelevant conditions *in vitro* may be challenging using standardized dissolution testing conditions. Researchers have used methods where the amount of dissolution medium is much reduced and size of the sample is close to the actual size of the nanofiber mat used *in vivo* [18,21,22,23]. Samples are typically collected at predetermined time intervals and analyzed by ultraviolet-visible (UV-VIS) spectrophotometry or high performance liquid chromatography (HPLC).

The biorelevant conditions applied for the drug release studies depend on the exact problem and site of application, and may vary. Hydrogels have been widely used as DDSs for topical applications; however, hydrogels may also provide for a simplistic wound model onto which drug may be released followed by drug diffusion into the hydrogel matrix. Diffusion is one of the major transport mechanisms in the wound [16,24], although swelling and erosion may also play a role depending on the formulation. For some electrospun nanofibers, the release rate has also been explained by desorption of the embedded drug from nanopores in the fibers or from the outer surface of the fibers in contact with the water bath [25]. In addition to the actual testing, simulations have been performed and models proposed that enable prediction of the drug release behavior of the electrospun fiber mats [16,26]. These models enable the design of mats with certain structures in order to achieve a desired drug release kinetics [12].

Recently, a fully automated fiber-optics based dissolution testing systems for *in situ* monitoring of drug release from electrospun fiber mats was proposed [27]. The direct ultraviolet (UV) measurement of dissolved drug within dissolution medium provided the dissolution profile in real-time. UV imaging technology has emerged in pharmaceutical analysis [28]. It has found use for the characterization of different pharmaceutical dosage forms, including monitoring drug release from capsules [29], patches [30], and hydrogels [31,32]. Spatially resolved absorbance values are measured facilitating monitoring of concentrations and concentration gradients by UV imaging, and thereby providing the potential for new insights to the drug dissolution and release processes through real-time monitoring of swelling, precipitation, diffusion, and partitioning phenomena [31,33,34]. Drug release from electrospun fiber mats into hydrogel system has not been investigated before and was of interest within the present study.

In addition to using physical methods in drug release studies, genetically engineered whole-cell bioreporters enable to obtain valuable information during drug release studies. A few of the main advantages of using bioreporters are that they provide physiologically relevant data by measuring only biologically available fraction of the chemical. A typical bioreporter consists of a biological recognition element (i.e., sensor), a transducer, and a reporter protein. Test chemical binds to the sensor element, a transducer initiates the production of the reporter protein, and a signal is produced [35,36]. In parallel to the drug release information, such methods give direct information about the bioactivity of the developed antimicrobial DDSs.

The aim of the current study was to test and compare different drug release model systems for the characterization of electrospun nano- and microfiber antibacterial drug-loaded mats. We studied two different polymeric compositions—polycaprolactone (PCL) alone or in combination with polyethylene oxide (PEO)—with the model antibacterial drug chloramphenicol (CAM). Interestingly, we have previously shown that although these fiber mats with different carrier polymers have different drug release behavior according to the dissolution test results, their antibacterial activity was rather similar in a disc diffusion assay [22]. Therefore, in order to understand drug release from electrospun polymeric fiber mats better, novel characterization methods were acquired in order to elucidate and rationalize drug release behavior. In the current study, the drug release from electrospun fiber mats into buffer solution and agar hydrogel was investigated using HPLC, UV-VIS spectrophotometry, and bacterial bioreporters responding to the antibacterial drug CAM. UV imaging was used for the first time to monitor real-time the drug release and diffusion from electrospun fiber mats into agarose hydrogel. Antibacterial activity testing of the CAM containing fiber mats was performed using disc diffusion assay in order to shed light on the correlation between the drug release and antibacterial activity and, hence, the intended use of electrospun mats as local antibacterial DDSs for wound infections.

## 2. Materials and Methods 

### 2.1. Materials, Bacteria and Release Media

*Drugs, polymers, supplies*. The antibacterial agent chloramphenicol (CAM) was used as a model active pharmaceutical ingredient. CAM, hydrophobic carrier polymer polycaprolactone (PCL, Mw ≈ 80,000), hydrophilic carrier polymer polyethylene oxide (PEO) (Mw ≈ 900,000), and all analytical grade reagents were purchased from Sigma-Aldrich Inc. (Darmstadt, Germany). Ampicillin sodium salt and anhydrous d-glucose used for the bacterial bioreporter preparation were obtained from Carl Roth GmbH + Co. (Karsruhe, Germany) and Fisher Scientific (Waltham, MA, USA), respectively. Type I agarose was purchased from Sigma-Aldrich (St. Louis, MO, USA). Agar hydrogel was prepared using Lennox lysogeny broth (LB) agar (Difco Laboratories, Detroit, MI, USA). FavorPrep Plasmid DNA Extraction Mini Kit and FavorPrep(TM) GEL/PCR Purification Mini Kit was purchased from Favorgen Biotech Corp. (Changzhi Township, Pingtung, Taiwan).

*Bacteria.* Bacteria (*Staphylococcus aureus* DSM No.: 2569) were obtained from Leibniz Institute DSMZ-German Collection of Microorganisms and Cell Cultures. *Escherichia coli* MG1655 strain [37] was used for biosensor construction. Cloning was performed in *E. coli* strain DH5α [38].

*Buffers and agarose hydrogel for drug release testing by UV imaging*. For the preparation of the phosphate buffer solution used as a dissolution medium, sodium dihydrogen phosphate dihydrate (NaH_2_PO_4_ · 2H_2_O, Merck, Darmstadt, Germany) was dissolved in distilled water. The pH of the solution (67 mM phosphate buffer) was adjusted to pH 7.40 using 5M sodium hydroxide solution. The 0.5% (*w*/*V*) agarose hydrogel was prepared by dissolving type I agarose in an appropriate volume of phosphate buffer kept at 98 °C for 45 min in a water bath. The gels were cast in the quartz cells, and allowed to settle for 30 min prior to commencing the release experiments.

*Buffers and agar hydrogels for drug release and antibacterial activity testing*. Drug release studies were conducted using phosphate buffered saline (1x PBS). Lennox lysogeny broth (LB) agar with a concentration of 1.5% (*w*/*V*) (*S. aureous*) and MOPS minimal medium [39] with 1.5% (*w*/*V*) agar (*E. coli*) were used for drug diffusion and antibacterial activity testing with bacteria.

### 2.2. Preparation and Characterization Methods

*Preparation of electrospinning solutions and fiber mats*. Fiber mats were prepared using an ESR200RD robotized electrospinning system (NanoNC, Seoul, Republic of Korea). Fiber mats with different compositions were prepared in order to provide different drug release kinetics. The exact compositions of the solutions used for electrospinning and electrospinning conditions are shown in Table 1.

A mixture of chloroform:methanol (3:1) (*V*/*V*) was used as a solvent system for the preparation of PCL and PCL/PEO systems, and a total of 10 mL was electrospun using rotation (20 rpm) and moving stage (speed 25 mm/min, distance 140 mm). For the preparation of the electrospinning solution, the polymers were dissolved in the solvent system under stirring overnight. The desired CAM concentration in the fibers was 4% (dry solid state %) and CAM was added together with the polymer into the solvent system immediately after the preparation. The electrospun fiber mats were collected onto aluminum foil and put into ziploc bags. The samples were kept in a desiccator at 0% relative humidity above silica gel to avoid humidity induced changes in the mats.

*Morphology and solid state characterization of electrospun fiber mats.* Morphology and diameter of electrospun fiber mats were investigated using scanning electron microscopy (SEM). Samples were mounted on aluminum stubs and magnetron-sputter coated with 3 nm gold layer in argon atmosphere prior to microscopy. Solid state characterization of the electrospun fiber mats and drug-loaded fiber mats was performed as described previously using attenuated total reflection-Fourier transform infrared (ATR-FTIR) spectroscopy (IRPrestige-21 spectrophotometer (Shimadzu Corp., Kyoto, Japan) with Specac Golden Gate Single Reflection ATR crystal (Specac Ltd., Orpington, UK) and verified with X-ray diffraction (XRD) (D8 Advance, Bruker AXS GmbH, Karlsruhe, Germany) [22]. The thickness of the fiber mats was verified using a Precision-Micrometer 533.501 (Scala Messzeuge GmbH, Dettingen, Germany) with the resolution of 0.01 mm. The thickness of the mats was 0.07 ± 0.01 mm for the PCL fiber (0.05 ± 0.01 mm with CAM) and 0.08 ± 0.01 mm for the PCL/PEO fiber (0.08 ± 0.01 mm with CAM) mats. 

*Drug loading and distribution in fiber mats.* High performance liquid chromatography (HPLC) (Shimadzu Prominence HPLC with LC20, PDA detector SPD-M2QA, controlled by LC Solution software (1.21 SP1 Shimadzu); Shimadzu Europa GmbH, Duisburg, Germany) was used to determine the CAM concentration in the electrospun fiber mats and to evaluate its distribution uniformity throughout the fiber mats. Analyses were performed according to the official European Pharmacopoeia method for a related substance CAM sodium succinate. Briefly, CAM-loaded fiber samples were cut into 1 cm^2^ pieces, weighed, and dissolved in chloroform and methanol (3:1 *V*/*V*). The HPLC measurements were performed using an octadecylsilyl column (Phenomenex, Luna C18(2), 250 × 4.6 mm, 5 μm). The flow rate was 1.0 mL/min, and injection volume was 20 μL. The mobile phase consisted of 2% phosphoric acid R, methanol R and water R in the volume ratio 5:40:55. A wavelength of 275 nm was used. 

### 2.3. Drug Release Studies 

#### 2.3.1. Drug Release Testing into Buffer Solution and Agar Hydrogel by UV-VIS Spectrophotometry and HPLC

Drug release to buffer solution and agar hydrogel was investigated using UV-VIS spectrophotometry and HPLC, respectively. 

*Release to phosphate buffer solution:* The *in vitro* drug release of CAM from electrospun PCL and PCL/PEO fiber mats was carried out as described previously [22]; however, a more frequent sampling protocol was used. Briefly, 4 cm^2^ samples (*N* = 3) cut from the mats were weighed and placed into 20 mL of 1× PBS (pH 7.4) at 37 °C in 50 mL plastic tubes. The tubes were put into a dissolution apparatus vessel (Dissolution system 2100, Distek Inc., North Brunswick, NJ, USA) containing water maintained at 37 °C using rotation (paddle system, 100 rpm). Aliquots of 2 mL were removed and replaced with the same amount of 1× PBS at set time points. The aliquots were analyzed using UV-spectrophotometry (Shimadzu UV-1800, Shimadzu Europa GmbH, Duisburg, Germany) at 278 nm.

*Release to agar hydrogel:* The amount of drug released into agar plates was investigated by sampling different zones of the agar (illustrated with a figure in Section 3.4). Pieces of fiber mat (PCL/CAM and PCL/PEO/CAM fiber mat discs, with a diameter of 1 cm) were weighed, put onto pre-warmed LB agar plates, kept at 37 °C, and removed at set time points. Zones of the agar were cut out, the agar sample was put into ethanol (96%) and sonicated for 15 min. This extraction process was repeated twice and the obtained ethanol solutions were combined. The vials with ethanol solutions were left under a fume hood without caps, for the ethanol to evaporate. The residues left in the vials were dissolved in 1.5 mL of ethanol (96%) and the amount of CAM analyzed with HPLC. In the present study, the limit of detection for CAM was 1 µg/mL. Triplicate measurements were performed. The extraction efficacy was tested separately confirming that two times extraction resulted in 100% efficacy (Appendix B, Table A1). 

#### 2.3.2. UV Imaging for Drug Release Monitoring in Hydrogel 

Complementary to the traditional HPLC method, an Actipix D200 Large Area Imager (Paraytec Ltd., York, England) controlled by Actipix D200 acquisition software ver. 3.1.7.4 was used to image the release of CAM from PCL and PCL/PEO fiber mats. These experiments were performed in a heating cabinet from Edmund Bühler TH30 (Bodelshausen, Germany) set to 37 °C. Imaging was performed at four alternating wavelengths: 525 nm, 280 nm, 255 nm, and 214 nm. Images for each wavelength were recorded at a frequency of 0.125 s^−1^ for the release experiments as well as the standard curve. The imaging area (28 × 28 mm^2^; pixel size 13.8 µm^2^) encompassed three quartz cells (Pion Inc., UK; 62 mm × 4 mm × 7 mm (L × H × W)), allowing three measurements to be performed simultaneously. The fibers were cut to fit the inner dimensions of the quartz cells (7.0 mm in width and 4.0 mm in height). The fibers were positioned perpendicular to the imaging direction in contact with the agarose gel. The fibers were backed by silicone plugs to ensure good contact with the gel and correct alignment. Parafilm was used to seal the quartz cells preventing evaporation of water from the gels. The release of CAM from the fibers was imaged for 3 h at 37 °C. Each imaging experiment allowed measurements of two CAM-containing fibers and one blank fiber (control). The positioning of the fibers in the imaging system (top, middle or bottom row) was randomized.

*Standard Curve for quantification by UV imaging.* A CAM stock solution (5 mM) in phosphate buffer was used to make the dilutions for the standard curve in 0.5% (*w*/*V*) agarose gel. These were made by mixing 1.5 mL 1% (*w*/*V*) agarose in phosphate buffer with a defined volume of CAM solution and phosphate buffer to obtain 3 mL of the mixture. The 1% (*w*/*V*) agarose solution and the phosphate buffer were both heated in water bath (98 °C) to facilitate mixing leading to a homogeneous mixture. The gels were cast in the quartz cells, and allowed to settle for 30 min prior to the experiments. As reference, a 0.5% (*w*/*V*) agarose gel in phosphate buffer without CAM was used.

### 2.4. Antibacterial Activity Studies—Drug Release and Effect on Bacterial Growth

#### 2.4.1. Antibacterial Activity Testing

The antibacterial activity of released CAM on agar plates was investigated at different time points mimicking the drug diffusion tests into agarose hydrogel during UV imaging studies. Overnight culture (20 h) of *S. aureus* DSM No.: 2569 was grown from DMSO stock (100 uL to 3 mL of LB). Preparation of all bacterial DMSO stocks used in the present study is described in Appendix C. The culture was diluted to optical density (OD) 0.05 in LB and 100 μL was plated onto pre-warmed LB agar plates (1.5% (*w*/*V*)). PCL/CAM and PCL/PEO/CAM fiber discs, and a positive CAM filter paper control were applied onto each plate. At specific time points, the discs were removed and the LB plates were incubated at 37 °C for 24 h prior to measurement of the inhibition zones.

#### 2.4.2. Bioreporter Plasmid and Strain Preparation 

All cloning was performed using CPEC cloning method [40]. Plasmid vector backbone was low-copy pSC101 plasmid. *Timer* reporter gene in plasmid pSC101-Ptet-Timer [41] was replaced with two fluorescent reporter genes *GFPmut2* [42] and *mScarlet-I* [43]. In order to increase the expression of the green fluorescence protein (GFP) during antibiotic stress additional stress-inducible dnaK1 promoter (PdnaK1) originating from *E. coli* MG1655 genomic DNA was added upstream of the tet-promoter (Ptet). In addition, kanamycin resistance gene *kanR* was replaced with ampicillin resistance gene *ampR.* In order to reduce the expression of *ampR* resulting from reverse direction transcription initiation from PdnaK1, additional rrnB T2 terminator was added between *ampR* and PdnaK1.

In order to construct the ribosomal stalling reporter plasmid pSC101-CAM-bioreporter transcription attenuation-based regulatory *trpL2Ala* region together with a terminator and constitutive T5 promoter from plasmid, pRFPCER-TrpL2A [44] was inserted between *GFPmut2* and *mScarlet-I* ribosomal binding site. mRNA from the reporter gene *mScarlet-I,* and therefore, red fluorescence is only produced when ribosomal stalling occurs, e.g., due to CAM presence (Appendix D). 

CPEC products were transformed into *E. coli* DH5α and plasmids were purified using FavorPrep Plasmid DNA Extraction Mini Kit. All plasmids were verified by sequencing. Purified bioreporter plasmid pSC101-CAM-bioreporter was transformed into *E. coli* MG1655 chemical competent cells via heat shock. The transformants were selected on ampicillin (100 μg/mL) containing LB-agar plates after overnight incubation. The full nucleotide sequence of the pSC101-CAM-bioreporter plasmid is provided as Appendix A in GeneBank (gb) file format.

#### 2.4.3. Bioreporter Disc Diffusion Assay

For bacterial bioreporter disc diffusion assay agar plates with defined MOPS minimal medium [39] supplemented with 0.4% (*w*/*V*) glucose as the carbon source and 1.5% (*w*/*V*) agar were prepared in sterile conditions by measuring 20 mL of warm agar medium per plate, and plates were dried for 30 min under laminar flow hood.

Bioreporter strain DMSO stock was thawed, diluted 20× into sterile 1× PBS and 75 μL was plated on each minimal plate. A sterile cotton bud dipped into 1× PBS was used to spread the cells evenly. Plates were left to incubate at 37 °C for 10 h. After incubation, the weighted fiber mats (PCL/PEO/CAM and PCL/CAM) were added to each plate. Individual plates were first scanned with the Amersham Typhoon scanner (GE Healthcare Europe GmbH, Freiburg, Germany) (pixel size 100 μm; green fluorescence: 488 nm laser, 525BP20 filter, PMT voltage 352V; red fluorescence: 532 nm laser, 570BP20 filter, PMT voltage 621V) after adding the mats and re-scanned every hour for 6 h. Scan time for each plate was approximately 6 min. The plates were incubated at 37 °C between the scans.

### 2.5. Data Analysis

Data are given as average ± standard deviation (SD), unless stated otherwise. Data were analyzed and figures plotted using MS Excel 2017 and/or 2016, GraphPad Prism 7 ver. 7.04 or OriginPro 8.5. Statistical analysis was performed by two-tailed Student’s *t*-test assuming unequal variances (*p* < 0.05) where applicable.

The SDI Data analysis software ver. 2.0.60624 (Paraytec Ltd., York, England) was used to analyze the recordings made on the Actipix D200 system. The CAM molar absorption coefficient determined from the standard curve was used to calculate CAM concentrations in the gels. The analysis was performed at 255 nm, because the linear range of the standard curve covered the absorbance values encountered in the release experiments in contrast to 280 nm. A 6.25 mm wide zone starting from the fiber-gel interface protruding into the gel in the CAM transport direction was defined. For each pixel column, the absorbance was averaged, converted to concentration and plotted as a function of distance from the fiber mat to attain concentration-distance profiles. From the concentration-distance profiles, the area under curve (AUC) was calculated to determine the total amount of CAM released from the fibers. 

The inhibition zones free of bacterial growth (diameters, mm) were determined using ImageJ software [45] program version 1.52n. Tests were run at least in triplicate. ImageJ software was also used for obtaining numerical values of fluorescent zones from disc diffusion assay images. Green and red fluorescence images were analyzed separately. 0.8 mm wide lines were chosen as regions of interest for analysis. The 1.5 cm long line was drawn starting from the fiber mat and plot profiles of grey values for these regions were recorded. 

## 3. Results and Discussion

### 3.1. Preparation and Characterization of Electrospun Fiber Mats

In order to test the suitability of the UV imaging technique and different drug release model systems for monitoring drug release from electrospun fiber mats, different mats were electrospun by varying the polymers (hydrophilic PEO vs. hydrophobic PCL) and incorporating the antibacterial drug CAM into the fibers (Table 1). The preparation and morphological and physicochemical characterization of these electrospun antibacterial CAM-loaded fibers has been performed previously [22]. As shown previously, the prepared PCL and PCL/CAM mats consisted of nanofibers within the average size range from 370 to 496 nm (SD ± 339 nm), whereas PCL/PEO and PCL/PEO/CAM mats were in the micrometer size range with an average diameter of 2.9 µm (SD ± 1.1 µm). Solid state transformation from crystalline CAM to amorphous CAM was confirmed with drug-loaded electrospun fiber mats (data not shown). In agreement with previous findings [22], CAM was homogeneously distributed within the electrospun fiber mats (data not shown) and CAM content matched with the theoretical values (Table 2).

### 3.2. Drug Release into Buffer Measured by UV-VIS Spectrophotometry

Initially, traditional dissolution testing into buffer solution was performed and CAM concentrations were determined using UV-VIS spectrophotometry. The PCL/CAM and PCL/PEO/CAM mats are different in terms of their wettability and swelling properties. PCL/PEO/CAM mats are more hydrophilic and swell when exposed to an aqueous medium [22]. The PCL/CAM mats are more hydrophobic, although the presence of CAM tends to increase the wetting of the mat and provide access for the buffer to enter the fibers. Thus, PCL/CAM fiber mats provided the expected and desired prolonged CAM release whilst PCL/PEO/CAM mats due to the hydrophilic nature of PEO exhibited a faster drug release in buffer solution (Figure 1).

Frequent sampling revealed a significant CAM burst release (up to 15 min of release testing) from both fiber mats. There were only minor differences in triplicate measurements and in the behavior of the fiber mats verifying the reproducibility of the measurements. In a recent study, the free drug (terbinafine hydrochloride) on the surface of fibers was removed after rinsing in distilled water and the amount of released drug quantified in a wound dressing-skin model utilizing filter paper as a matrix [21]. The rinsing procedure most likely removed the drug burst release. In the present study, no pretreatment of the mats was performed and the mats were analyzed directly. It is likely that if burst released amounts were removed the drug release would be even more different between the mats and most of CAM would be removed from PCL/PEO/CAM fibers mats during the pretreatment. For understanding drug release and *in vivo* activity relationships, the mats should not be pretreated; however, such pretreatment might be important if *in vitro* tests are performed to illustrate only the differences between the fiber mats in respect of their prolonged drug release.

### 3.3. Antibacterial Activity of Electrospun Fiber Mats

Despite the different morphologies and behavior (e.g., swelling, drug release into buffer), statistically significant differences in the inhibition zones on agar plate between the PCL/CAM and PCL/PEO/CAM fiber mats were not observed during previous antibacterial activity testing [22]. In the present study, it was of interest to investigate further how the drug is released from the electrospun mat into a gel which more closely resembles the wound matrix (e.g., agar and agarose hydrogels) and how this translates into antibacterial effect. Hydrogels have more similar hydrodynamic conditions to wound tissue as compared to aqueous solutions and thus provide more biorelevant testing option. We developed a modified disc diffusion assay, where the mats were physically removed from the surface of the solid growth medium at specified time points and the antibacterial effect of the released drug on model bacteria *S. aureus* DSM No.: 2569 was determined by measuring the size of the inhibition zones after 24 h. The faster CAM release from PCL/PEO/CAM mats compared to the PCL/CAM mats observed in buffer solution (Figure 1) correlated with larger inhibition zones (Figure 2). The differences were larger at the earlier time points. Control filter paper impregnated with CAM confirmed that the wetting of the sample is the major triggering factor for further drug release and diffusion.

### 3.4. Drug Release into Agar Hydrogel Measured by HPLC

The extent of CAM release from the mats and diffusion into the agar hydrogel was quantified using HPLC upon extraction of CAM from the agar hydrogel. The agar on the plates was divided into five concentric circular zones (Figure 3A) and the CAM concentration in each zone was determined. The experimental design is detailed in the Materials and Methods and Appendix B (Table A1 and Table A2). The amount of drug released from electrospun mats to the agar plates at different time points at 37 °C is summarized in Figure 3. 

The drug diffusion patterns for PCL/CAM and PCL/PEO/CAM fiber mats were qualitatively similar (Figure 3). These findings are consistent with the disc diffusion assay results (see for Figure 2). It is clear that most of the drug is present in the 1 cm diameter section (zone 1) right below the fiber mat at the early time points of the release experiment. As time passed, less drug was recovered in the inner circle (zone 1) and relatively more in zone 2 (larger circle around the mat). CAM release from PCL/PEO/CAM fiber mat was fast followed by slower diffusion further into the agar hydrogel (Figure 3B). Interestingly, after 120 min, the drug was not detectable in the 3rd zone. After 24 h (1440 min), the CAM had reached the outer circle (Appendix B, Table A2). PCL/CAM fiber mats on the other hand showed that less drug was released within the same time period compared to the PCL/PEO/CAM fiber mat (Figure 3C). The CAM concentration distribution differences are visible in Figure 3, mainly for zone 1 but to some extent also for zone 2. This may explain why it is not always possible to detect differences in the inhibition zones between different fiber mats in a disc diffusion assay although the drug may be released differently from the delivery vehicle and, therefore, lead to different antibacterial efficacy *in vivo* [46].

### 3.5. Detecting CAM Release into Agar Hydrogel with Bioreporter Strain

In addition to chemical and physical methods for determining the CAM concentrations during the release, it is also possible to use bacterial bioassays. Hence, simultaneously to the antibacterial effects, the drug release can be monitored. We genetically engineered reporter bacteria (*E. coli* MG1655) to produce dose-dependent quantifiable green and red fluorescent signals in the presence of antibacterial drug CAM. The CAM-bioreporter has GFP as a control protein for expression and a red mScarlet-I as a reporter protein. Exact working mechanism of the bacterial bioreporter is provided in Appendix D. In the presence of CAM GFP signal (green) will be reduced due to protein synthesis inhibition, and mScarlet-I signal (red) will increase due to transcription continuation as a result of ribosomal stalling in the transcription attenuation system. Therefore, the CAM release and diffusion can be illustrated in different time points (selected time points 60, 180, and 360 min) using fluorescence data (Figure 4).

It is clearly seen that bacterial growth is inhibited close to the fiber mats (inhibition zones surrounding the mats) where CAM concentrations are the highest, which matches with the agar diffusion test results (Figure 2 and Figure 3). Fluorescent bacteria surrounding the inhibition zones reveal the distance from the mat where the CAM levels above the minimum inhibitory concentration (MIC) can still be detected. Compared to the agar hydrogel diffusion tests (Figure 2 and Figure 3), bacterial bioreporter study results on agar hydrogel did not reveal large differences between the PCL/CAM and PCL/PEO/CAM fiber mats with the respect of released drug amounts and its effect on bacteria (Figure 4C). Growth inhibition was very slightly more pronounced in PCL/PEO/CAM fiber mat (fast release) after 60 min of incubation; however, this difference was statistically insignificant and disappears in later time points (180 min). In later time points (6 h) red fluorescence detected from bacteria enabled determining the CAM concentrations even below MIC (in sub-MIC concentrations) (Figure 4D,E). After 6 h of incubation, the peak of the reporter protein signal is located further (approximately 0.75 cm) from the mat in case of PCL/PEO/CAM fiber mat (fast release) (Figure 4D) compared from PCL/CAM fiber mat (slow release; located approximately 0.60 cm) (Figure 4E). This indicates that effective CAM concentration was achieved on larger area (at a further distance from the fiber mat) with PCL/PEO/CAM fiber mat, although the difference between the two fiber mats was minor. Most likely, this is due to the fact that most of the differences between the two different electrospun fiber mats can only be seen in early time points which cannot be distinguished using fluorescent bacteria in these settings.

### 3.6. Drug Release into Agarose Hydrogel Using UV Imaging

The assessment of CAM release by cutting of zones from agar plates followed by extraction and analysis of the drug is both a destructive and laborious sampling procedure. The fast release of CAM cannot be monitored using bioreporter bacteria that require time for growth and detectable signal production. Therefore, UV imaging was investigated as a potentially less labor intensive and more robust (reproducible/repeatable) approach for monitoring CAM release from the electrospun fiber mats. An additional benefit of the non-intrusive imaging method is the high spatial as well as temporal resolution offered. Due to a lack of transparency of the agar plates used in antibacterial activity tests, a hydrogel matrix based on agarose was applied instead. Moreover, the Actipix D200 Large Area Imager allowed imaging of three samples in parallel enabling comparison. Images visualizing the release and diffusion of CAM at different time points are shown in Figure 5A. 

Diffusion into the viscous agarose hydrogel resembles the conditions encountered both during *in vitro* disc diffusion testing (agar hydrogel) as well as *in vivo* where the drug has to be released from the electrospun fiber and diffuse into the wound. The UV imaging results reveal that PCL/CAM fibers provide slower drug release into the gel whereas PCL/PEO/CAM fibers release the drug faster (Figure 5A). The concentration-distance profiles shown in Figure 5B,C correspond to the images in Figure 5A. In addition, the average amount of CAM released was quantified based on the UV images (Figure 6). When comparing the two mats, differences were observed with respect to the total amount of released drug mainly at the early time points (Figure 6).

Only slight differences between the PCL/CAM and PCL/PEO/CAM fiber mats with respect to their drug release behavior were detected in average drug release profiles, although these were not statistically significant and can be considered as more of a tendency. 

### 3.7. Comparison between Different Drug Release Model Systems

Irrespective of the *in vitro* release model system (testing approach) employed, the release of CAM was found to be faster from the PCL/PEO fiber mats as compared to the PCL fiber mats (Figure 7). However, the extent to which the release differed tended to vary between the model systems.

Drug release into buffer revealed that after 30 min the PCL/PEO/CAM fibers had released nearly all CAM into the buffer, whereas only a minor amount (approximately 30%) of drug was released from PCL/CAM fiber mats (Figure 7). When the CAM release into hydrogel was monitored by agar disc diffusion and extraction method, the amount of CAM released from PCL/PEO/CAM fiber mat after 30 min was approximately 70% and the released CAM amount from PCL/CAM mat was similarly approximately 30% (with huge variability between 17–48% for individual PCL/CAM fiber mats). UV imaging, however, showed that the differences with respect to the amount of drug released into the agarose hydrogel between PCL/CAM and PCL/PEO/CAM fiber mats after 30 min were less pronounced as compared to the differences in CAM release into buffer (Figure 7). It was also seen that much less CAM was released and diffused to the region of analysis from both fiber mats (approximately 9% and 8% for PCL/PEO/CAM and PCL/CAM fiber mats, respectively) within the same time period. Most likely, less CAM is released from the fiber mats into the hydrogel according to the UV imaging due to the different geometries of the setup. In the UV imaging setup, diffusion is limited to one direction perpendicular to the mat, whereas diffusion in the agar setup may occur in three dimensions. The latter will favor a relatively larger release of CAM. We believe that the concentration may be more likely to build up at the interface in the UV imaging setup. As might be expected due to the similarity of the release matrices (hydrogel), the UV imaging drug release and diffusion profiles matched more with the drug release into agar hydrogel extraction (Figure 3, Figure 4 and Figure 5) than release and dissolution into buffer test (Figure 1). The contact and/or of the mats with the hydrogels may be different between the agar and agarose hydrogels. It is not known whether the agar hydrogel may contain components interacting, and thereby facilitating the release of CAM. Similarly to UV imaging, less pronounced differences between the two different electrospun CAM-loaded fiber mats were also observed with bacterial bioreporters (Figure 4C). It is due to the fact that the drug has to diffuse away from the mat before it will be detected (UV imaging and bacterial bioreporters), whereas in the agar hydrogel extraction method, the area under the fiber mat will be included into the total released and detected drug amount. 

The differences with respect to CAM release from the mats observed using the *in vitro* release testing model systems are related to the different properties of the release media (volume, agitation, and viscosity), its capability to penetrate into the fiber mat, the sample size of fiber mats, effect of shaking, and geometry of the setup. It is believed that drug diffusion and wetting of the samples were the rate-limiting steps for the drug release from fiber mat into agar and agarose hydrogels. The release into buffer solution, however, was more affected by the hydrodynamics of the buffer measurements. For the different fiber mats agitation led to increased erosion and/or disintegration of the PCL/PEO/CAM fiber mats as compared to the PCL/CAM fiber mats.

In terms of repeatability the model systems and analytical methods were all comparable. It can be seen that larger variations in drug release between replicates were detected for PCL/CAM fiber mats compared to PCL/PEO/CAM fiber mats. This was seen with both drug release model systems (buffer vs. hydrogel) as well as different techniques (UV-VIS spectrophotometry, HPLC, UV imaging, and bacterial bioreporters). This might be explained by the different wettability of the PCL/CAM fiber mats compared to PCL/PEO/CAM fiber mats as discussed previously for PCL and PCL-drug loaded fiber mats [15]. The hydrophilicity of the PCL/PEO/CAM fiber mats makes them wet more homogeneously, whereas for PCL/CAM fiber mats the sample-to-sample variations in wettability may also cause variability in the drug release behavior. 

Compared to agar diffusion assay (extraction of CAM from hydrogel), the UV imaging was easier to conduct. Although for both these methods the size of sample, the release environment (hydrogel) and static conditions resembled the *in vitro* antibacterial activity testing conditions. Bacterial bioreporter study enabled to monitor the CAM release into agar hydrogel during diffusion in later time points where even sub-MIC concentration of CAM was determined based on the green and red fluorescence intensity. The method has the advantage of revealing the released antibacterial drug effect directly on bacteria.

For polymer fiber mats as wound matrices it is favorable to have the burst effect. The extent of the burst release is often associated with device geometry, surface characteristics of host material, heterogeneous distribution of drug within the polymer matrix, intrinsic dissolution rate of drug, and heterogeneity of matrices [16]. From the drug release experiments we see that the burst release is smaller with the hydrogels for hydrophilic PCL/PEO/CAM and hydrophobic PCL/CAM fiber mats compared to the buffer solution (Figure 7). Larger differences in the burst release between the samples were observed with hydrophobic PCL/CAM fiber mats (Figure 7). For the drug release to occur from the PCL fiber mat, the medium needs to penetrate into the electrospun fiber mat to cause the diffusion of the drug to the exterior. Initially, the diffusion is affected by the electrospun mat composition and the fiber characteristics, as for PCL/PEO/CAM fiber mats the release was also affected by swelling and dissolution of PEO. As a second step, the diffusion further into the wound and reaction with the surrounding tissue and environment is important [47]. It is clear that the buffer release method shows higher burst release as compared to the *in vitro* antibacterial activity testing. Sadri et al. have measured tetracycline release in a dialysis bag mimicking the human skin-like conditions and shown that the burst release effect is reduced most likely due to the diffusion through the bag, which modified the release [48]. The gels *in vitro* and the wound tissue *in vivo* can be envisioned to provide a diffusion barrier or matrix minimizing burst release effects. Local transport mechanisms determine the volume in which the drug is distributed; thus, understanding these mechanisms is essential for optimizing the effectiveness of local delivery [47]. The *in vitro* methods that enable to most closely mimic the *in vivo* conditions are likely to enable accurate predictions of the drug release and transport in the wound environment. Moreover, it is important to understand the effect on bacteria. Therefore, the response of the bacteria to the active antibiotic concentration has to be measured directly. Here, we have shown that genetically engineered bioreporter systems can be used for this purpose. 

## 4. Summary and Conclusions

All tested drug release model systems enabled to study the drug release from electrospun fiber mats. UV imaging provided faster, non-intrusive real-time information on chloramphenicol (CAM) release and diffusion with higher spatial and temporal resolution as compared to the release methods relying on release from fiber mats directly into buffer solution as well as to the agar plates followed by destructive sampling. The data were similar in terms of sensitivity in detecting differences between the electrospun fiber mat compositions in comparison to the more laborious drug release testing setups. Furthermore, the UV imaging and release into hydrogel method appeared to mimic the *in vitro* antibacterial activity testing conditions to a large degree, while release into buffer solution overestimated the burst release effects. 

Advanced antibacterial disc diffusion assays enabled to distinguish small differences in the antibacterial activity of the differently designed electrospun fiber mats, these slight differences were also visualized using UV imaging as well as modified drug release and diffusion into hydrogel model systems. These results suggest that diffusional resistance to drug release observed in the hydrogel based systems may emulate the *in vivo* conditions of the wound matrix.

## Figures and Tables

**Figure 1 pharmaceutics-11-00487-f001:**
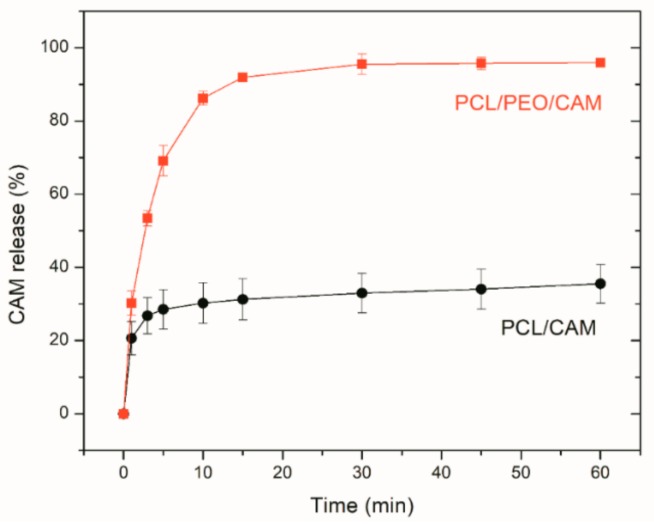
Relative release of CAM from PCL/PEO/CAM (∎) and PCL/CAM (●) fiber mats into phosphate buffered saline at pH 7.40 and 37 °C. Data are averages ± SD of at least triplicate samples. Analyses performed using UV-VIS spectrophotometry (reference is made to Table 1 for fiber composition and preparation conditions).

**Figure 2 pharmaceutics-11-00487-f002:**
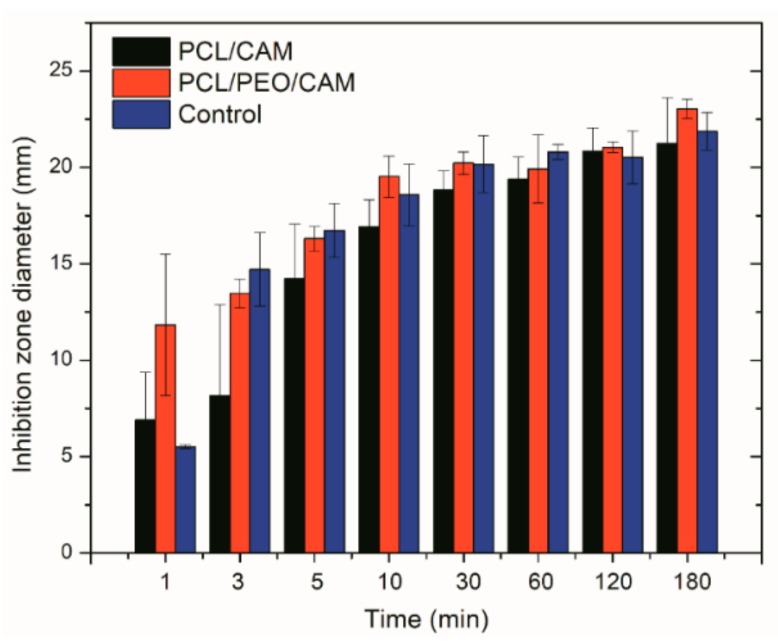
Inhibition zone diameters measured after 24 h on agar plates. X-axis indicates exposure time of discs (PCL/PEO/CAM or PCL/CAM fiber mats) onto the hydrogel. *S. aureus* DSM No.: 2569 was used for the study. Filter paper wetted with CAM solution and dried (same CAM concentration) served as a control. Key: CAM, chloramphenicol; PCL, polycaprolactone; PEO, polyethylene oxide. Data are averages ± SD of at least triplicate samples.

**Figure 3 pharmaceutics-11-00487-f003:**
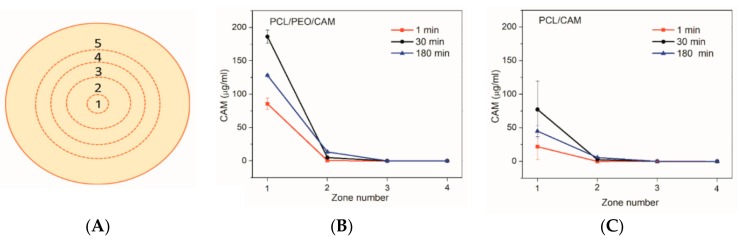
(**A**) Schematic illustrating the division of the agar plate into zones. Zones 1-5 are numbered starting from the inner circle. Zone diameters: 1.0 cm, 3.0 cm, 4.4 cm, 5.8 cm, and 8.6 cm. Concentrations of released CAM (detection limit of 1 µg/mL) from PCL/PEO/CAM (**B**) and PCL/CAM fiber mats (**C**) into 1.5% (*w*/*V*) agar hydrogel at 37 °C in different time points (1 min, 30 min, 180 min) and into different zones (1–4). Error bars represent standard deviation, *N* = 3. Key: CAM, chloramphenicol; PCL, polycaprolactone; PEO, polyethylene oxide.

**Figure 4 pharmaceutics-11-00487-f004:**
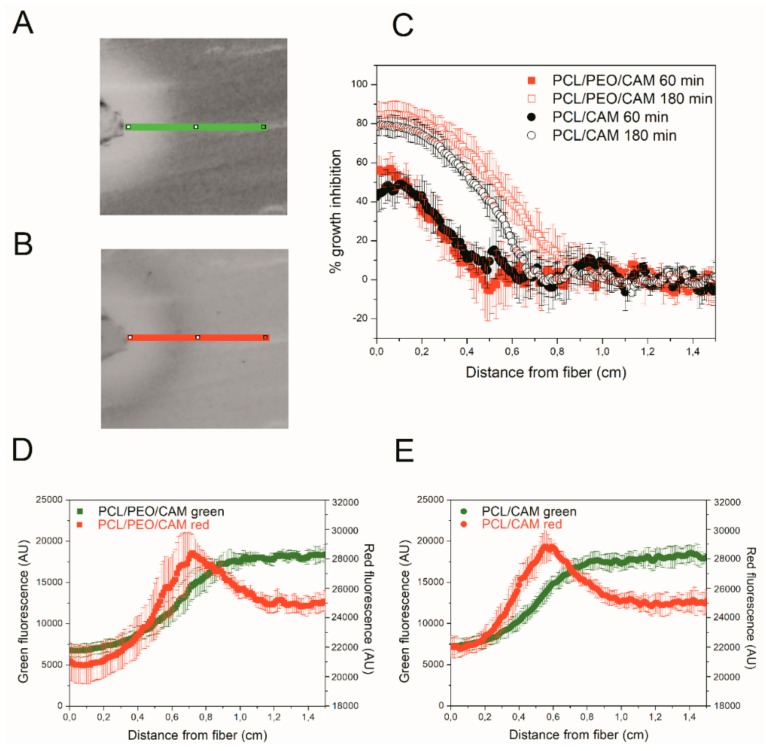
(**A**,**B**). Shows the analyzed region of interest (0.8 mm × 1.5 cm) of bioreporter chloramphenicol (CAM)-containing fiber disc diffusion assay from green fluorescence (**A**) and red fluorescence (**B**) scan images. Green fluorescence images in different time points (60 min and 180 min) allow estimating bacterial growth inhibition due to CAM released from PCL/PEO/CAM and PCL/CAM fiber mats (**C**). Combined green and red fluorescent figures at 6 h reveal the fluorescence levels that can be correlated with the released CAM from PCL/PEO/CAM (**D**) and PCL/CAM fiber mats (**E**). Data are presented as average of three experiments (± SD). Key: CAM, chloramphenicol; PCL, polycaprolactone; PEO, polyethylene oxide.

**Figure 5 pharmaceutics-11-00487-f005:**
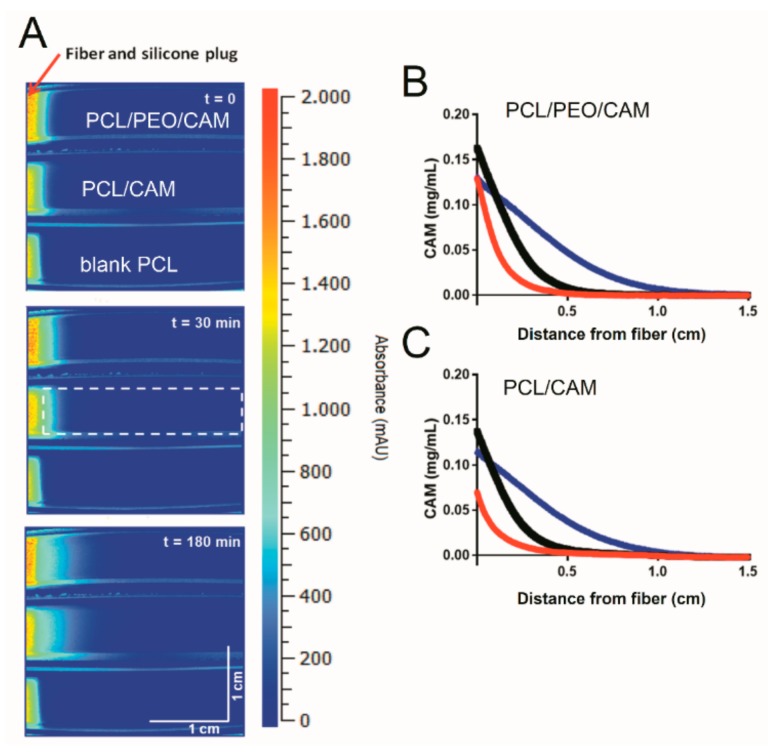
(**A**) Representative images of CAM release from fibers and diffusion into 0.5% (*w*/*V*) agarose hydrogel at 37 °C. Top is PCL/PEO/CAM, middle is PCL/CAM and bottom is PCL/Blank fibers (control). The zone used for the quantification of the CAM absorbance is shown in the middle row at 30 min. Representative concentration-distance profiles for PCL/PEO/CAM (**B**) and PCL/CAM (**C**) fibers diffusing into 0.5% (*w*/*V*) agarose hydrogel at 37 °C. The red, black and blue line is for 0 min (immediately after starting the experiment), 30 min and 180 min, respectively. Key: CAM, chloramphenicol; PCL, polycaprolactone; PEO, polyethylene oxide.

**Figure 6 pharmaceutics-11-00487-f006:**
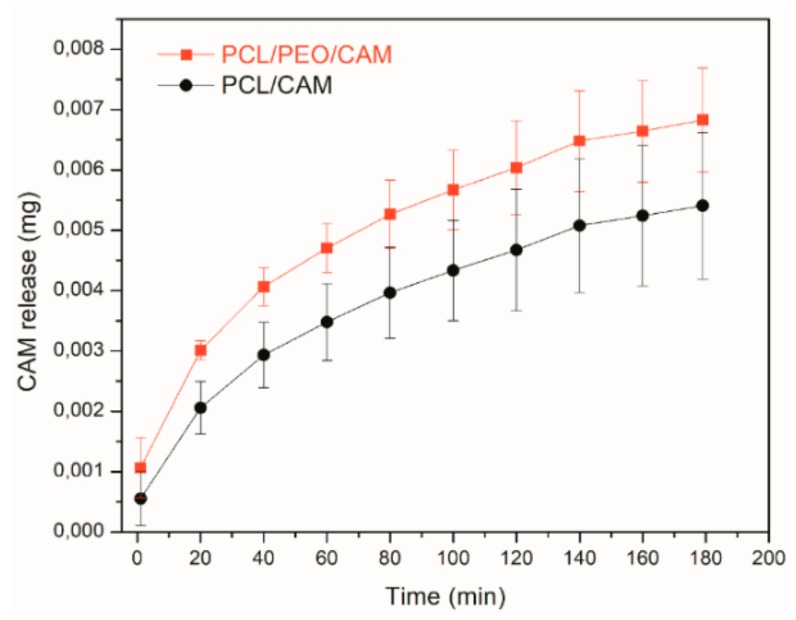
Amount of CAM released from PCL/PEO/CAM (∎) and PCL/CAM (●) fiber mats into 0.5% (*w*/*V*) agarose hydrogel at 37 °C based on UV imaging experiment data. Error bars represent one standard deviation, *N* = 5. Key: CAM, chloramphenicol; PCL, polycaprolactone; PEO, polyethylene oxide.

**Figure 7 pharmaceutics-11-00487-f007:**
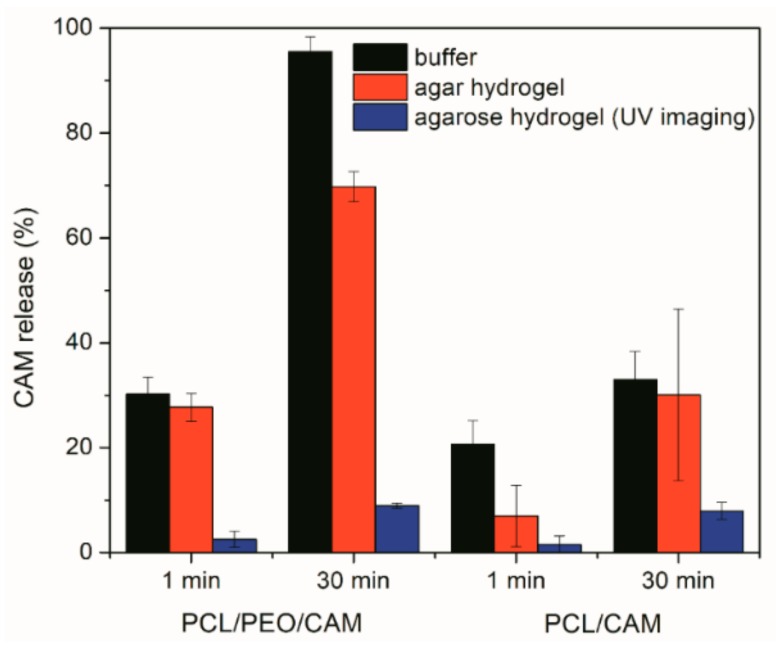
Amount of CAM released from PCL/PEO/CAM and PCL/CAM fiber mats into phosphate buffered saline at pH 7.40 and 37 °C and 1.5% (*w*/*V*) agar hydrogel at 37 °C and 0.5% (*w*/*V*) agarose hydrogel at 37 °C. Data are averages ± SD of at least triplicate samples. Analyzes performed using UV-VIS spectrophotometry (CAM concentration in a buffer solution), HPLC (CAM concentration extracted from hydrogel) and UV imaging (CAM concentration within hydrogel) (reference is made to Table 1 for fiber composition and preparation conditions). Key: CAM, chloramphenicol; PCL, polycaprolactone; PEO, polyethylene oxide.

**Table 1 pharmaceutics-11-00487-t001:** Composition of the electrospun formulations and electrospinning parameters.

Formulations	Materials/Polymer	Materials/Solvent	Distance (cm)	Flow Rate (ml/h)	Applied Voltage (kV)
PCL	PCL 12.5% (*w*/*V*), CONTROL	Chloroform:methanol (3:1 *V*/*V*)	14	1.0	9
PCL/CAM	PCL 12.5% (*w*/*V*) + CAM (4% *w*/*w*, solid state)	Chloroform:methanol (3:1 *V*/*V*)	14	1.0	9
PCL/PEO	PCL 10% + PEO 2% (*w*/*V*), CONTROL	Chloroform:methanol (3:1 *V*/*V*)	17	2.5	12
PCL/PEO/CAM	PCL 10% (*w*/*V*) + PEO 2% (*w*/*V*) + CAM (4% *w*/*w*, solid state)	Chloroform:methanol (3:1 *V*/*V*)	17	2.5	12

Key: CAM, chloramphenicol; CONTROL, formulation without CAM; PCL, polycaprolactone; PEO, polyethylene oxide.

**Table 2 pharmaceutics-11-00487-t002:** Average CAM concentrations within the electrospun fiber mats (*N* = 3–7). Formulation compositions and electrospinning parameters are shown in Table 1.

Electrospun Fiber Formulations	Theoretical CAM Content/%	Measured CAM Content/% ± SD
**PCL/CAM**	4	4.0 ± 0.2
**PCL/PEO/CAM**	4	3.8 ± 0.3

Key: CAM, chloramphenicol; SD, standard deviation.

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
