# Peer review of "Monitoring of Antimicrobial Drug Chloramphenicol Release from Electrospun Nano- and Microfiber Mats Using UV Imaging and Bacterial Bioreporters"

_pharmaceutics, 2019, doi:10.3390/pharmaceutics11090487_

Round 1

Reviewer 1 Report

This is an interesting work about monitoring of antimicrobial drug chloramphenicol release from electrospun nano- and microfiber mats using UV imaging and bacterial bioreporters. The characterization is complete and the science is solid. Therefore, I think it should be qualified for Pharmaceutics. However, I still have some minor concerns about this work.

(1) Some images presented in the manuscript are not clear enough. For example, Figure 3 and Figure 4(D)-4(E).

(2) The error analysis of this work is not properly handled. For example, in table 2, "4.0 ± 0.17" should be "4.0 ± 0.2".

(3) In the introduction, the authors have briefly introduced the electrospinning nanofibers. However, the authors have not introduced the applications of electrospun nanofibers, such as air filtration (Macromolecular Materials and Engineering, 2017, 302(1), 1600353-1600380; Macromolecular Materials and Engineering, 2018, 1800336-1800354), antibacterial (ACS Appl. Mater. Interfaces,2019,11,13,12880-12889) and drug delivery (Journal of Materials Chemistry B 2019, 7, 709–729). I think it will make the manuscript appeal to broad interest including those applications in the introduction.

Based on the above concerns, I suggest a minor revision.

Author Response

Response to Reviewer 1 Comments

Thank you for the comments, the raised questions and the responses can be found below:

Comment: This is an interesting work about monitoring of antimicrobial drug chloramphenicol release from electrospun nano- and microfiber mats using UV imaging and bacterial bioreporters. The characterization is complete and the science is solid. Therefore, I think it should be qualified for Pharmaceutics. However, I still have some minor concerns about this work.

Point 1: Some images presented in the manuscript are not clear enough. For example, Figure 3 and Figure 4(D)-4(E).

Response 1: As suggested by the Reviewer, we have checked all the figures in the manuscript and increased their quality. New figures will be sent together with the revised manuscript.

Point 2: The error analysis of this work is not properly handled. For example, in table 2, "4.0 ± 0.17" should be "4.0 ± 0.2".

Response 2: We have corrected this.

Point 3: In the introduction, the authors have briefly introduced the electrospinning nanofibers. However, the authors have not introduced the applications of electrospun nanofibers, such as air filtration (Macromolecular Materials and Engineering, 2017, 302(1), 1600353-1600380; Macromolecular Materials and Engineering, 2018, 1800336-1800354), antibacterial (ACS Appl. Mater. Interfaces,2019,11,13,12880-12889) and drug delivery (Journal of Materials Chemistry B 2019, 7, 709–729). I think it will make the manuscript appeal to broad interest including those applications in the introduction.

Response 3: We have modified the introduction section and added the suggested reference about the drug delivery. However, we prefer not to widen the introduction of the manuscript to cover applications besides biomedical applications. We believe that in the context of pharmaceutics and drug delivery other applications such as air filtration is of limited relevance.

We have modified the introduction by adding new references (suggested by the Reviewer):

Drug delivery and tissue engineering:

Gao, S.; Tang, G.; Hua, D.; Xiong, R.; Han, J.; Jiang, S.; Zhang, Q.; Huang, C. Stimuli-responsive bio-based polymeric systems and their applications. J. Mater. Chem. B 2019, 7, 709–729.

Sill, T.J.; von Recum, H.A. Electrospinning: Applications in drug delivery and tissue engineering. Biomaterials 2008, 29, 1989–2006

Torres-Martinez, E.J.; Cornejo Bravo, J.M.; Serrano Medina, A.; Pérez González, G.L.; Villarreal Gómez, L.J. A Summary of Electrospun Nanofibers as Drug Delivery System: Drugs Loaded and Biopolymers Used as Matrices. Curr. Drug Deliv. 2018, 15, 1360–1374.

Ye, K.; Kuang, H.; You, Z.; Morsi, Y.; Mo, X. Electrospun Nanofibers for Tissue Engineering with Drug Loading and Release. Pharmaceutics 2019, 11.

The revised manuscript with new high quality figures can be also found in the attachment. In addition, we send the high quality figures and cover letter explaining made changes to the Editor.

Reviewer 2 Report

Please find below my comments:

Is there any difference in the morphology of PCL/CAM and PCL/PEO/CAM? I would guess that PEO will diffuse out once that the fibers are immersed in a buffer, leading to high porosity and thereby higher release rate compared to the PCL/CAM formulation. In my view, SEM images of the two formulations should be added to show the eventual difference in morphology.

Can the authors explain the choice of the hydrogels' material to mimic the skin?

The authors use throughout the text dissolution and release almost interchangeably which is confusing. Please check and use each term appropriately.

Author Response

Response to Reviewer 2 Comments

Thank you for the comments, the raised questions and the responses can be found below (the modified text from manuscript has been written in Italics and changes made in the original text is highlighted):

Point 1: Is there any difference in the morphology of PCL/CAM and PCL/PEO/CAM? I would guess that PEO will diffuse out once that the fibers are immersed in a buffer, leading to high porosity and thereby higher release rate compared to the PCL/CAM formulation. In my view, SEM images of the two formulations should be added to show the eventual difference in morphology.

Response 1: There are differences in the morphology of PCL/CAM and PCL/PEO/CAM fiber mats, as reported previously in Preem et al. [22]. SEM micrographs are indeed very important when considering the morphology of fiber mats and their performance. However, as we have already published the SEM micrographs of these fibers, we prefer not to duplicate this, but rather make comment on the morphology difference and include a reference to [22]. We have modified the sentences in the manuscript:

The preparation, morphological and physicochemical characterization of these electrospun antibacterial CAM-loaded fibers has been performed previously [22]. As shown previously, the prepared PCL and PCL/CAM mats consisted of nanofibers within the average size range from 370 to 496 nm (SD ± 339 nm), whereas PCL/PEO and PCL/PEO/CAM mats were in the micrometer size range with an average diameter of 2.9 µm (SD ± 1.1 µm).

Reference:

[22] Preem, L.; Mahmoudzadeh, M.; Putrinš, M.; Meos, A.; Laidmäe, I.; Romann, T.; Aruväli, J.; Härmas, R.; Koivuniemi, A.; Bunker, A.; et al. Interactions between Chloramphenicol, Carrier Polymers, and Bacteria–Implications for Designing Electrospun Drug Delivery Systems Countering Wound Infection. Mol. Pharm. 2017, 14, 4417–4430.

Point 2: Can the authors explain the choice of the hydrogels' material to mimic the skin?

Response 2: The selection of the hydrogels used was essentially based on three criteria. First, the increased similarity of hydrogels to wound tissue as compared to aqueous solutions due to changed hydrodynamic conditions (increased biorelevance), secondly the established use of agar in antibacterial testing made agar an obvious choice, and, third, the requirement of having a transparent matrix for UV imaging which makes agarose an ideal material (in contrast to agar). We explained shortly about the use of hydrogel as a release medium in the introduction and Results and Discussion sections (agar diffusion testing and UV imaging paragraphs), however in order to highlight this more we have modified the manuscript (page 9) as follows:

In the present study, it was of interest to investigate further, how the drug is released from the electrospun mat into a gel which more closely resembles the wound matrix (e.g., agar and agarose hydrogels) and how this translates into antibacterial effect. Hydrogels have more similar hydrodynamic conditions to wound tissue as compared to aqueous solutions and thus provide more biorelevant testing option.

Point 3: The authors use throughout the text dissolution and release almost interchangeably which is confusing. Please check and use each term appropriately.

Response 3: Thank you for this comment. We have checked the entire manuscript in order to distinguish the two terms and modified the manuscript accordingly. In case of electrospun drug-loaded fiber mats, we need to discuss both of these phenomena; how the drug is released and how it is dissolved within the medium.

The revised manuscript with high quality figures can be found in attachment. In addition high quality figures and cover letter explaining the made changes have been sent to the Editor.

Reviewer 3 Report

This manuscript reported drug release model system for the characterization of electrospun nano- and microfiber antibacterial drug loaded-mat using two different polymeric composition, PCL and PCL-PEO. The paper is well-written and interested, but some minor parts should be clarified. 

1. Fig.2 showed that filter paper is also similar antibacterial behavior. Then, what advantage of electrospun fiber mats we can expect?

2. In Fig.7, what is the difference between agar hydrogel and agrose hydrogel? why is CAM release pattern are different? Looks this is a key data of this project. the reason should be discussed more in detail.

3. What is the purpose to show bacterial bioreporter study? It is alternative method for drug release model or just comparison with UV-imaging? 

Author Response

Response to Reviewer 3 Comments

Thank you for the comments, the raised questions and responses can be found below:

Comment: This manuscript reported drug release model system for the characterization of electrospun nano- and microfiber antibacterial drug loaded-mat using two different polymeric compositions, PCL and PCL-PEO. The paper is well-written and interested, but some minor parts should be clarified.

Point 1: Fig.2 showed that filter paper is also similar antibacterial behavior. Then, what advantage of electrospun fiber mats we can expect?

Response 3: In this case, the filter paper was used as a control in order to compare the drug concentration effects on bacteria. Indeed, it was interesting to observe that filter paper with drug adsorbed to the surface (or sorbed within the structure) may also provide some kind of prolonged release due to the diffusion into the gel. An antibacterial activity was expected as the chloramphenicol concentration selected should be sufficiently high to have an antibacterial effect.

The advantage of the electrospun fiber mats is that they are made of materials (polymers) acting to control the drug release and support the wound healing. In addition, electrospun fibers provide high drug loading and encapsulation efficiency. As this is the development phase, the modification of drug release using different materials is the key point in order to understand the structure and activity relationships. It is believed that it is easier to control drug release from the electrospun structures rather than from drug adsorbed on the surface of a material (although nanofiber mats may also have some drug present on the surface of the mat/fibers, which most likely will be released immediately upon exposure to the aqueous medium). The placement of drug within the fiber depends on the polymer, drug, solvent as well as electrospinning conditions.

Point 2: In Fig.7, what is the difference between agar hydrogel and agrose hydrogel? why is CAM release pattern are different? Looks this is a key data of this project. the reason should be discussed more in detail.

Response 2: Agar and agarose hydrogels were used with different drug release measurements, the first (agar) was used for agar diffusion and extraction experiments and agarose was used for the UV imaging experiments. Agar hydrogel was not used for UV imaging due to the interference from the background as explained in the manuscript. CAM release profiles are most likely different because different experimental geometries applied. The UV imaging setup allowing diffusion in only one direction would be expected to lead to a lower amount of drug release as compared to the agar assay. We have elaborated on the possible causes for differences observed in the manuscript on pages 15-17; adding some more explanations in order to improve the discussion and highlight the observations.

Point 3: What is the purpose to show bacterial bioreporter study? It is alternative method for drug release model or just comparison with UV-imaging?

Response 3: We used the bacterial bioreporter study as an alternative method to monitor drug release from such electrospun fiber systems. In the present study all the methods investigated were compared using the same electrospun fiber mats. This was helpful in shedding light on the limitations and advantages of different drug release model systems and their suitability for characterisation of antimicrobial drug-loaded matrices.

The revised manuscript with high quality figures can be found in attachment. In addition, we have submitted the high quality figures and cover letter explaining the made changes to the Editor.
